# Tourists' Perceived Attitudes toward the Famous Terraced Agricultural Cultural Heritage Landscape in China

**Xiaopiao Yang** [1,2], **Yuluan Zhao** [1,2,*], **Jia Zhao** [1], **Chao Shi** [1] **and Bailu Deng** [3]

1   School of Geography and Environmental Science, Guizhou Normal University, No. 116, Baoshan North Road, Guiyang 550001, China
2   The State Key Laboratory Incubation Base for Karst Mountain Ecology Environment of Guizhou Province, No. 180, Baoshan North Road, Guiyang 550001, China
3   School of Horticulture and Landscape Architecture, Southwest University, No. 2, Tiansheng Road, Beibei, Chongqing 400715, China
*   Correspondence: zyl_009@126.com

**Abstract:** Terraces are the major vehicle for agricultural activities in mountainous areas and are an important component of the agro-cultural heritage landscape. This work explores tourists' perceived attitudes toward, and characteristics of terraced agro-cultural heritage landscapes based on online web travel notes. A framework of visitor perception types of terraced agricultural cultural heritage landscapes was constructed, and each type was analyzed in a targeted manner. The results obtained can provide a reference for the conservation of heritage farming culture and the development of strategies to improve landscape quality for such sites. This study used crawler software to collect online travelogue data from 3991 notes by visitors to seven note-worthy terraced agro-cultural heritage sites in China and used the ROST Content Mining 6 tool to analyze high-frequency feature words, semantic networks, and sentiment distribution and ten-dency. We found that the tourist perception of the diversity of terraced agro-cultural heritage landscape is rich, with a high overall evaluation. The tourists' perceptions focused on four elements: landscape, ecology, culture, and service. They were more likely to have a high perception of the landscape than service, which in turn was higher than culture and ecology. The emotional tendency of tourists' perceptions is mainly highly positive and neutral, and negative emotions account for a lower proportion and are mostly mild.

**Keywords:** terraced fields; agricultural cultural heritage; landscape; perception; travel notes; tourist

## 1. Introduction

China has the largest number of globally important agro-cultural heritage sites in the world [1], and the agro-cultural system based on terraced fields occupies an important proportion of China's globally important agricultural heritage system and has gradually become its core component [2]. The terraced fields are an objective reflection of China's historical agro-culture and ancient farming wisdom [3]; they are a unique land management system and agricultural landscape formed by long-term co-evolution and adaptation with the countryside [4]. Terraced fields have various functions, such as maintaining regional ecological balance, ensuring regional food security, providing high-quality landscape resources, and protecting the life community of mountains, rivers, forests, farmlands, lakes, grasses, and biodiversity [5,6]. However, due to the migration of rural labor from mountainous areas and the inefficiency of income from agricultural industry, terraced fields are facing prominent problems such as a sharp reduction of landscape coverage, functional degradation, unsustainable inheritance of agricultural culture, and insufficient industrial integration and development [7–9]. It is undeniable that, with the general development of rural tourism in China, there are new opportunities for the development of terraced agro-cultural heritage, which has become an important resource supporting agricultural landscape tourism. Exploring new paths coupling agro-tourism development

in terraced agro-cultural heritage tourism sites and tourists' attitudes toward tourism in such landscapes can provide important ideas for guiding, enhancing, and forecasting the development of agro-tourism in China [10].

Research on tourism development in rural tourism sites based on participatory appraisal and questionnaire interview methods has yielded mature theoretical results [11,12]. With the increasing application of Internet technology in scientific research fields such as agriculture, tourism, and rural development [13–15], analysis of tourism website visitor data has become an important way for researchers to judge the development of tourism destinations [16–18]. Travel notes are the textual embodiment of tourist attitudes and perceptions of the landscape and the specific tourist destinations visited during their travel [19]. Semantic analysis is widely used for travel text analysis to determine the root cause of tourism service quality problems [20], the formation mechanism of tourists' perceived image, and tourism demand forecasting [21–23]. The application of tourist evaluation is becoming increasingly mature. The vast majority of studies have been conducted on individual destinations, and there are no multi-perspective studies focused on the same landscape type. From a macro-regional perspective, there are limitations in conducting tourist perception evaluation studies on a single tourism site. For the terraced agro-cultural heritage landscape, the geographical variability of different terraced landscapes in terms of natural geographical environment, socio-economic development level, and cultural background is an objective fact; it is not convincing to represent the results of tourists' perception of China's terraced agro-cultural heritage landscape as a whole based on attitudes toward a particular site. Previous studies on individual terraced agricultural landscape objects thus lack universality and extensiveness, and they do not provide a strong indication of tourism attitudes and affective tendencies toward the terraced agro-cultural heritage landscape. There has also been a lack of systematic generalization of tourists' perceptions of the terraced agro-cultural heritage landscape. Based on this, this article takes the Hani terraces in Yunnan, the Longji terraces in Guangxi, the Ziquejie terraces in Hunan, the Jiabang terraces in Guizhou, and other famous terraces in China as the object of study, obtaining the web tour evaluations of the research subjects with the help of web crawler software, and uses the integrated travelogue network text analysis method to excavate the content of tourists' perceptions of this landscape and proposes the perception types of tourist in landscape, ecology, culture, and service based on these perspectives, to find the optimization path of the terraced agricultural cultural heritage landscape by analyzing the perception results of each type. The results reveal the conservation and development problems faced by these sites and provide a reference for addressing their urgent practical needs.

## 2. Literature Review

### 2.1. The Terraced Landscape Carries Important Agricultural Cultural Heritage Values

The terraced field is a type of arable land created by humans to adapt agricultural production activities that are not conducive to crop-growing, such as mountains, slopes, and hilly terrain [24]. They mainly distribute in tropical and subtropical mountains, temperate deserts, arid and semi-arid hilly areas, and even alpine fragile mountains, throughout Asia, Europe, America, Africa, and other continents [25]. Terraced fields were used only by peasants to carry out agricultural production activities during the agrarian and feudal societies. Historically, it is the most direct objective embodiment of farming civilization in modern society and has witnessed the development of human civilization for thousands of years [26]. The Food and Agriculture Organization of the United Nations (FAO) defines globally important agricultural cultural heritage (GIAHS) as "unique land-use systems and agricultural landscapes resulting from the long-term synergistic evolution and dynamic adaptation of the countryside to its environment, which are rich in biodiversity and which meet the needs of local socio-economic and cultural development and contribute to the sustainable development of the region [27]". In 2010, the first international conference on terraced landscapes was held in Yunnan, China. The conference established the International Terraced Landscape Alliance (ITLA) and adopted the Honghe Declaration on the

Conservation and Development of Terraced Fields [28]. The conference proposed that "the protection and development of terraced civilization is the common responsibility of the whole society", and the participating experts generally agreed that the typical terraced landscape should receive extensive attention from the academic community as agricultural cultural heritage [29]. Among the globally important agricultural cultural heritage announced by FAO, 18 agricultural cultural heritages in China have been selected for the list [2], and nearly 50% of them are closely related to the terraced landscape. It shows that the terraced landscape carries important agricultural cultural heritage values.

### 2.2. Utilization Pathways for the Terraced Fields, from Inefficient to Efficient

In terms of labor inputs to mountain agricultural activities and the benefits they generate, for most farmers, engaging in terracing activities brings them losses rather than profits [30]. However, the interest of urban people in rural tourism sites and agricultural landscapes will gradually increase [31], and the label of "agriculture, history, culture and landscape" of the terraced fields makes them an important resource for rural revitalization [32,33] when urbanization and modernization reach a certain level. The phenomenon of terraces being abandoned due to the lack of agricultural labor and the difficulty of farming in mountainous areas is becoming more and more serious [34]. The disappearance of landscape features of terraced fields has caused the reduction in the quality of agricultural landscape, the decrease in the high added value of terraced agricultural cultural heritage, and the weakening of farmers' motivation to engage in agricultural activities [35]. Rural tourism sites based on terraced agricultural cultural heritage landscapes face serious tests of sustainability. In this background, whether the government, farmers and developers can effectively cooperate becomes the key to whether the terraces can realize the appreciation of agricultural industry. Therefore, in order to make the best use of the multifunctional advantages of the terraced landscape, to enhance the high added value of the terraced agricultural cultural heritage, and to find the path from inefficient to efficient utilization of the terraced agricultural cultural heritage, we conducted a study of tourists' perceived attitudes toward the terraced agricultural cultural heritage landscape, and hope that the results can provide theoretical references for realizing landscape conservation of terraced agricultural cultural heritage and improving the quality of life of farmers in terraced areas.

### 3. Materials and Methods

#### 3.1. Study Area

Terraces are built according to the mountainous terrain and are an agricultural landscape formed by the scientific and rational exploitation of the vertical structure of the mountain and people's interaction with the natural environment to provide arable land with high-quality soil and that can conserve water [36]. The mountainous area of southern China is wide, with an undulating terrain. The region has a subtropical monsoon climate with rain and heat in the same period, providing suitable conditions for rice growth. The terraced landscapes in China are mainly distributed in the southern area, south of the Huaihe River in the Qinling Mountains and east of the Qinghai–Tibet Plateau, concentrated in the Yunnan–Guizhou Plateau Mountains and some hilly provinces in the southwest, such as the Hani Terraces in Yunnan Province, the Jiabang Terraces in Guizhou Province, the Longji Terraces in Guangxi Province, the Ziquejie Terraces in Hunan Province, the Shangbao Terraces in Jiangxi Province, the Lianhe Terraces in Fujian Province, and the Yunhe Terraces in Zhejiang Province, which are among the most typical ones. Terraced fields not only have significance in maintaining regional food ecological security and stability but are also extremely precious agro-cultural landscapes, and the seven samples studied have formed an agricultural tourism industry based on these terraced fields, which has made important contributions to regional economic and social development.

### *3.2. Data and Data Sources*

This paper used the travel diaries posted by tourists on travel websites with famous Chinese terraced agro-cultural heritage sites as their destination as the main data source. Data collection was carried out on travel websites such as Ctrip, Mafengwo, and Tuniu using the web crawler software Octopus Collector. On the travel notes page, we searched for the famous domestic terraced agro-cultural heritage landscapes using keywords and crawled the travel text in the search results. The travel notes of each tourist were the evaluation unit for the summary and general processing of the text data. The obtained travel data could be entirely image based or there could be many images with few words; such samples cannot effectively express tourists' subjective feelings; after general processing, the primarily imaged-based or invalid travel notes with limited text were manually removed. A total of 3991 valid travel notes were obtained (Table 1), which included 994 in 2017, 1006 in 2018, 984 in 2019, 314 in 2020, 539 in 2021, and 154 in 2022, -approximately 17 million words.

**Table 1.** Distribution of travel notes collected from tourism websites.

| Attractions | Travel website | | | | |
|---|---|---|---|---|---|
| | Ctrip | Mafengwo | Tuniu | Lvmama | Qyer |
| Longji Terraces, Guangxi | 932 | 94 | 171 | 17 | 227 |
| Hani Terraces, Yunnan | 324 | 69 | 10 | 82 | 150 |
| Yunhe Terraces, Zhejiang | 267 | 55 | 36 | 8 | 34 |
| Lianhe Terraces, Fujian | 699 | 67 | 2 | 87 | 16 |
| Jiabang Terraces, Guizhou | 205 | 95 | 18 | 6 | 52 |
| Shangbao Terraces, Jiangxi | 32 | 61 | 8 | 0 | 47 |
| Ziquejie Terraces, Hunan | 49 | 60 | 4 | 0 | 7 |

Note: Ctrip, Mafengwo, Tuniu, Lvmama and Qyer are famous travel websites in China.

### *3.3. Methods*

Web text analysis to study tourists' travel notes is an important method for understanding tourists' perceptions of the image of a tourist destination; it is also an important reference for guiding the improvement of tourism service quality at destinations [37]. Web text analysis can be used to analyze the specific content of documents systematically, objectively, and quantitatively; it has been widely used in tourism and human geography and has yielded fruitful results [38,39]. The current network text analysis method was based on the ROST Content Mining 6 (ROST CM6) text analysis tool developed by Professor Shenyang and his team at Wuhan University in China [40], which has been widely used in humanities and social science research. After the general processing of the text, the software analyzes the textual material in terms of word frequency, social and semantic networks, sentiment, and clustering, among other elements. Based on 3391 valid tourist travel notes about terraced cultural landscapes crawled from well-known domestic tourism websites, this paper used general processing, word segmentation, high-frequency feature word extraction, social and semantic network analysis, and sentiment analysis to explore the prominent image of terraced agro-cultural heritage from the tourist perspective to understand their emotional tendencies, explore their perception of such heritage sites, construct a framework of visitor perception types of terraced agricultural cultural heritage landscapes, and propose relevant landscape enhancement strategies.

## 4. Results

### *4.1. Word Frequency Feature and LECS Perception Type Analysis*

Based on the general processing of the travel notes by the ROST CM6 software, secondary processing was applied in combination with the word division results, which involved redefining feature words that were not effectively separated out using custom word lists, using human–computer interaction to filter invalid words and to obtain feature words matching the evaluation of the target landscape. Rice terraces, as conservation

objects, contain three parts: the terrace landscape, ecology, and culture [41]. Tourists' perception of the terraced cultural landscape has multiple types unified with the whole, rather than an intuitive perception of a single element. The sources of people's feelings in the terraced landscape environment include natural landscapes such as mountains, rivers, and lakes; cultural landscapes, such as rural customs and folklore; systematic ecological landscapes, such as biodiversity; material and intangible cultural landscapes, such as characteristic buildings; and tourism services. Based on the in-depth understanding of the sample travel notes, we sought to fully excavate the connotations of agro-culture within the local folk culture, local customs, and tourists' emotional and psychological processes. Based on the word frequency analysis of the first 500 feature words, the tourists' perceptions of terraced agro-cultural heritage were classified into four types of perceptions through the difference analysis of characteristic words (Table 2): landscape, ecology, culture, and service.

**Table 2.** Perception type, target, and main semantic feature words.

| Perception Type | Main Perception Target | Semantic Aware Feature Words |
|---|---|---|
| Landscape | Natural existing and man-made landscapes, including terraced fields, forests, villages, squares, buildings, and other landscape elements | Terraced fields, landscape, nature, buildings, view, beautiful scenery, sunrise, observation deck, shooting, spectacular, stockade, paradise, clouds, fairyland, golden, meandering, layered, high and low, idyllic, shape, line, curve, ink painting, patchwork, row upon row, picturesque scenery, beautiful, nice, canyon, canal, irrigation, forest, etc. |
| Ecology | Ecological elements in the overall landscape, including biodiversity, environmental quality, and vegetation density | Emerald green, environment, ecology, air, green, fresh, original ecology, green mountains and waters, beautiful mountains and rivers, good environment, dense vegetation, soil erosion, diverse plants, diversity, rich species, beautiful environment and sustainable, etc. |
| Culture | Traditional cultural landscapes in the region, including traditional song and dance performances, local ethnic costumes, festival customs and other intangible cultures, as well as traditional ethnic architecture and other material cultures | Performance, singing and dancing, history and culture, ethnic minorities, customs, enthusiasm, ancient, heritage, humanities, ancient architecture, human landscape, simplicity, rice wine, national culture, national costumes, traditional culture, etc. |
| Service | Service experience generated by tourists during their travel, including hotel accommodation, transportation, and convenience | Travel, transportation, cheap, gourmet, worth, snack, lively, safe, homestay, hotel, inn, hygiene, comfort, excitement, wonderful, satisfaction, expectation, warmth, moving, tourism product, over-development, experience, discount, feel, etc. |

The top 500 synonymous and near-synonymous feature words were merged based on perception type by, for example, combining feature words such as inn and homestay into hotel, merging view and landscape into scenery, and combining rural, countryside, and stockade into villages. The first 50 perceptual high-frequency words were taken for type recognition (Figure 1) to analyze tourists' perception preferences for the target landscape. The results showed that the most frequent feature words in the axes were mainly distributed in landscape perception and service perception, while the feature words of ecology perception and culture perception were concentrated at the bottom of the axes.

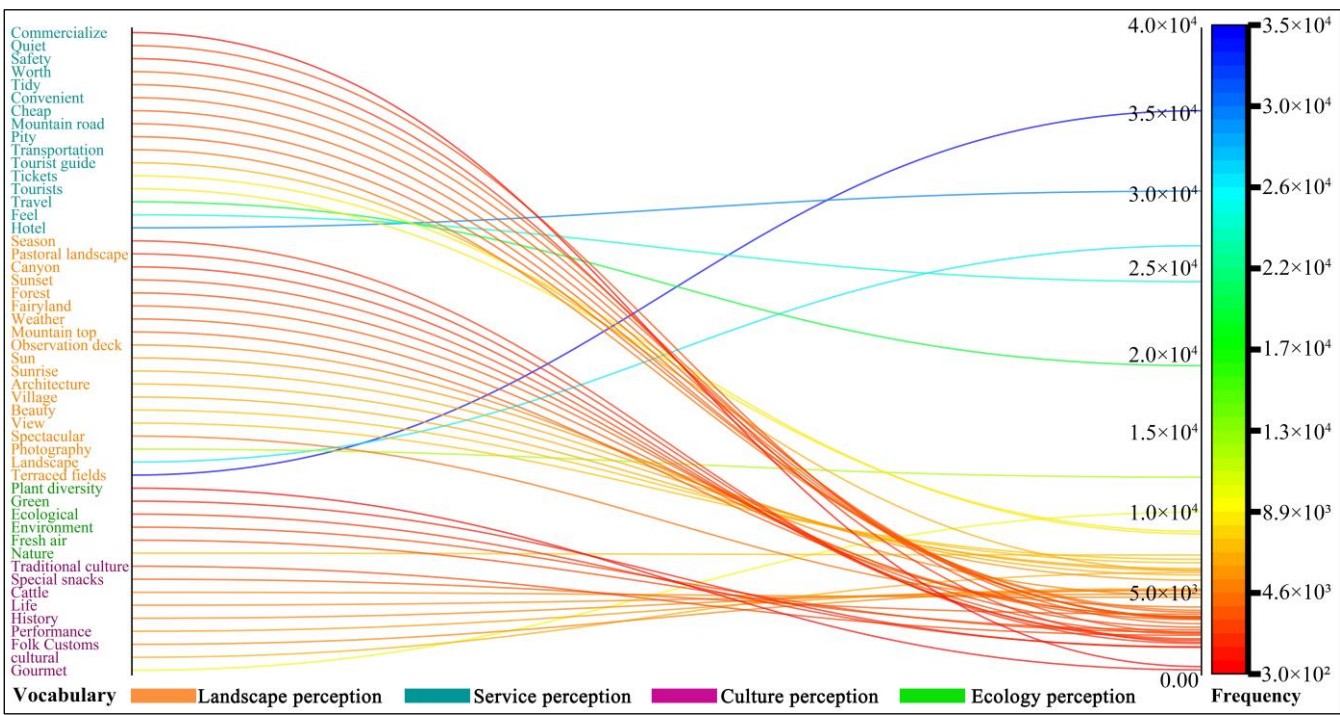

**Figure 1.** Top 50 high-frequency feature words (after merging synonyms).

In the tourist perception of the agro-cultural heritage of the terraced fields, the degree of the four types of perception can be ranked, from strongest to weakest, as landscape, service, culture, and ecology. The degree of landscape perception was the strongest, with a related word frequency of 148,141, accounting for 47.2%. This was followed by service perception, with a word frequency of 129,707, accounting for 41.2%. The third was cultural perception, with a word frequency of 49,818, accounting for 6.3%. The weakest degree of perception was for ecology, with a word frequency of only 16,793, accounting for 5.3% (Figure 2). The top five characteristic words—terraced fields (34,860 times), hotels (29,895 times), landscapes (26,528 times), feelings (24,611 times), and tourism (19,125 times)—were mainly based on the perception of landscape and service. There were significant differences and distributional weights for the perceptual preferences based on the four types. The results indicated that the objective visible, audible, and tactile landscapes and visitors' experiences in the course of tourism activities were perceived most strongly in the terraced agricultural cultural heritage, while non-objective presence in the scene, such as cultural, ecological, and other elements, were perceived less strongly. Therefore, it was appropriate for managers of terraced agricultural cultural heritage tourism sites to strengthen the management of the physical landscape, while integrating the intangible "cultural" and "ecological" attributes into specific visitor tourism programs, for example, in the terraced landscape with high perception, agricultural experience projects such as "sowing, rice planting, and harvesting" can be added to allow visitors to truly participate in agricultural activities, or cultural elements such as "special festivals" and "ethnic symbols" can be added to the hotel. On the one hand, it can improve the tourism economic benefits of the scenic spot. On the other hand, it can increase visitors' perception of the intangible landscape, and facilitate the inheritance of intangible culture and the formation of visitors' awareness of ecological protection.

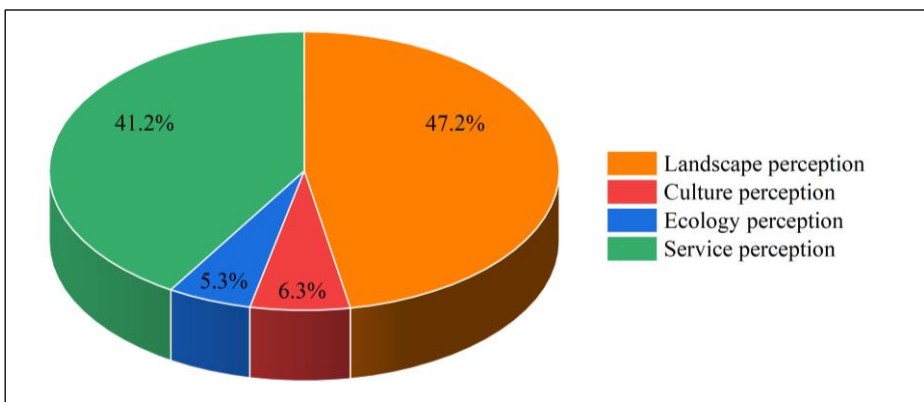

**Figure 2.** Distribution of degree of different perception types.

*4.2. Social and Semantic Network Analysis*

The co-occurrence matrix of high-frequency words indicated the strength of the connection between these words and explains the preference and intention of tourists' tourism activities. The word frequency analysis results showed that "terrace" was the core word with the highest frequency in the samples (34,860 occurrences). The co-occurrence matrix of tourists' attitudes toward terrace agro-cultural heritage landscape was generated based on the co-occurrence social network analysis, and the top 20 words co-occurring with terrace were extracted. After merging words with the same semantic meaning, the high-frequency co-occurring words appeared to be concentrated in aspects such as time, scenic spots, scenery, tourism, hotels—that is, they were mainly related to travel experiences such as commuting time, scenic landscape quality, and accommodation services (Figure 3). This showed the high importance of landscape quality and tourism experience in the tourist perception of terraced landscapes.

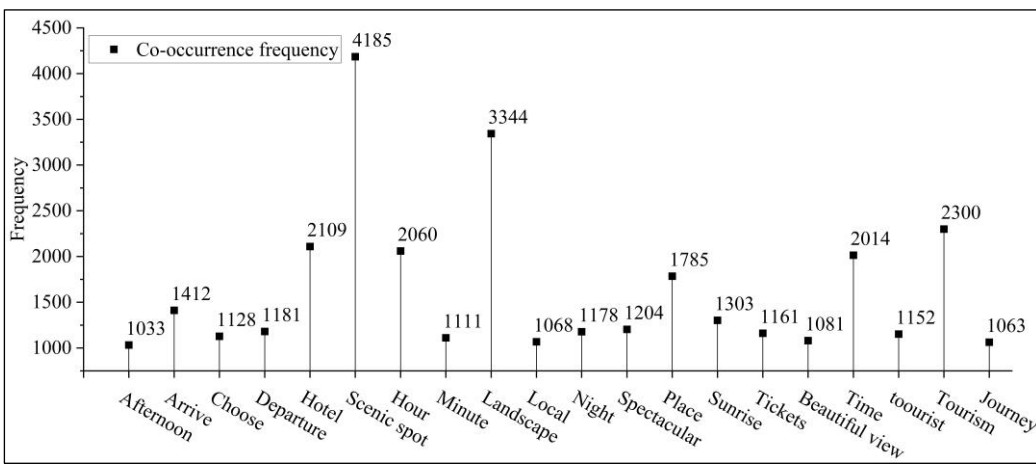

**Figure 3.** Top 20 words that co-occur with "terraced fields".

According to the graphs based on semantic network analysis (Figure 4), the graph with the words terraced fields, landscape, and tourism as the core had a clear hierarchical structure, and the structures from the core to the periphery are closely related. The concentrated results of tourists' perception of the agro-cultural heritage of terraced fields reflected the core status of landscape and service perception. Moreover, words such as time, trip, and ticket were significant in the semantic network. The analysis results were similar to the behavioral feature word frequency and high frequency word co-occurrence analyses, which also showed the importance that tourists attach to landscape optimization and service enhancement. These two elements should therefore be targeted in the development and enhancement of terraced agro-cultural heritage tourism sites. The results showed that

"terraces" as a core word was strongly associated with other elements. In the process of developing strategies for the enhancement and optimization of the overall landscape, the strategy makers and scenic developers need to coordinate the relationship between the elements. First, ensure that the quality and quantity of the "terraced landscape" are not reduced. At the same time, improve the comprehensive service level of scenic spots, increase the training of the service personnel of scenic spots in terms of service attitude and working ability to enhance the travel experience of tourists. The attention to "time, trip and tickets" suggested that managers need to focus on optimizing commuting times between attractions, and offering as many tour routes as possible for visitors to choose from.

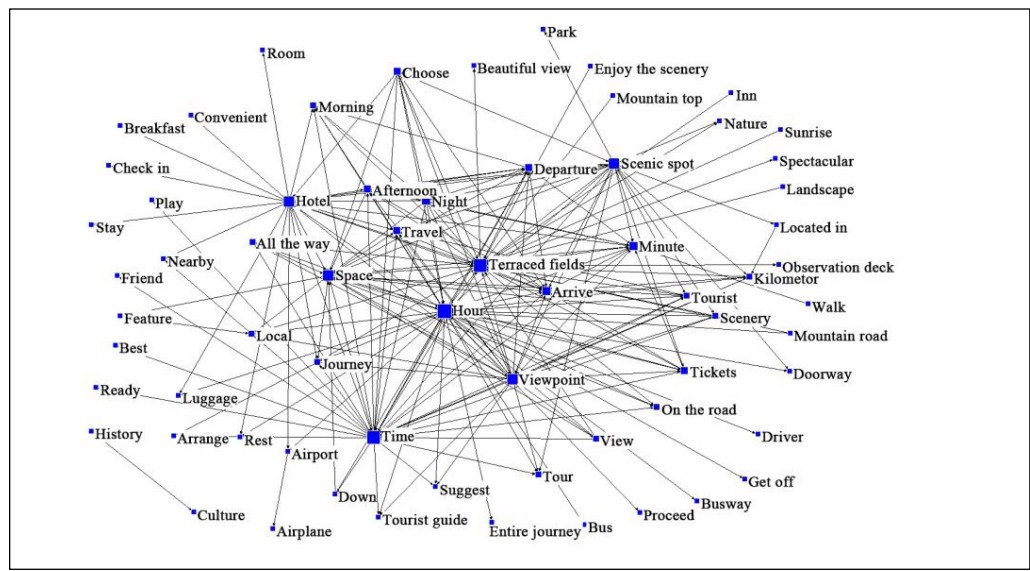

**Figure 4.** Semantic network diagram.

### 4.3. Sentiment Analysis

Sentiment analysis based on a sentiment dictionary was used to judge the tendency of the text (travel notes) according to three scales: positive, negative, and neutral [42,43]. This has guiding significance for which direction should be chosen for optimizing the sites, as well as which promotion strategy should be used. A travel note may contain perceptions and attitudes toward multiple landscapes simultaneously; to allow sentiment analysis results to cover a wide range of landscape objects in each travel note, instead of taking each travel note as the research unit, the evaluation was based on the entire travel note text and the emotional words recognized by the software in the emotional dictionary. According to the ROST CM6 sentiment analysis results, the emotional tendencies of tourists in China's famous terraced agro-cultural heritage landscapes were dominated by positive and neutral emotions (Table 3), accounting for 52.82% and 36.45%, respectively, of all responses. Negative emotions accounted for the lowest proportion, with only 10.73% of all responses. Emotions of all tendencies had different levels. ROST CM6 divides positive and negative emotions into three subsections: general, moderate, and severe; the analysis results showed that the degree of positive emotions was relatively balanced, with generally, moderately, and severely positive emotions accounting for 45.75%, 25.13%, and 29.12% of responses, respectively. The negative emotion segmentation was mainly concentrated in the mildest degree of negative (generally negative), which accounted for 78.95% of negative responses, while moderately negative emotions accounted for 18.89% of negative responses, and severely negative emotions accounted for only 2.16% of negative responses.

Table 3. Sentiment analysis of tourists' perception of terraced agro-cultural heritage.

| Emotion Category | Frequency | Proportion |
|---|---|---|
| Positive emotions: | 61,872 | 52.82% |
| Of which: | | |
| Generally positive (0–10): | 28,305 | 45.75% |
| Moderately positive (10–20): | 15,548 | 25.13% |
| Severely positive (>20): | 18,019 | 29.12% |
| Neutral emotions: | 42,698 | 36.45% |
| Negative emotions: | 12,567 | 10.73% |
| Of which: | | |
| Generally negative (−10 to 0): | 9440 | 78.95% |
| Moderately negative (−20 to −10): | 2259 | 18.89% |
| Severely negative (<−20): | 258 | 2.16% |

Note: Frequency represents the number of times emotional feature words appear in travel notes.

The emotional tendency of tourists' perception of the terraced agro-cultural heritage landscape thus appears to be dominated by positive emotions, with the lowest percentage of negative emotions, which were generally mild. The tourists' attitude is generally positive and focuses on the perception of the landscape and service in the tourism experience, including words such as beautiful, shock, national characteristics, and enthusiasm in the evaluations. There were few negative emotions in tourists' travel notes, and those that do appear mainly focus on the settlement landscape and tourism experience, the inconvenient transportation to the terraced sites, and the serious commercialization and over-development caused by the development of regional ethnic culture. The results of the sentiment analysis indicated that the terraced agro-cultural heritage landscape need to be improved around the "negative attitudes" of tourists, and particular attention should be paid to the optimization of "accessibility" and "commercialization". Meanwhile, improving the evaluation system, establishing a complaint platform, and adopting a systematic vertical supervision management using "employees-stores-attractions-scenic spots-government administration" will facilitate the overall development of the terraced agricultural cultural heritage landscape.

## 5. Discussion

Online travel notes are an important reference in guiding the travel decisions of potential tourists. Analyzing tourists' attitudes based on online travelogue texts, making good use of agricultural landscape resources, and coordinating the rational allocation of resources are effective measures to realize the development and utilization of landscape resources, as well as the formulation of a conservation strategy to promote rural revitalization [44,45]. The classification of LECS types perceived by tourists to terraced agro-cultural heritage sites was determined based on the overall characteristics of the landscape, and this can cover natural material elements such as forests, villages, terraced fields, and water systems, as well as non-material elements such as ecology, culture, and experience. Through the analysis of LECS perception types, tourists' preferences can be more comprehensively clarified, and the direction of tourism development can be optimized. Based on our analysis, we found that tourists pay special attention to landscape quality and service experience in the perception of seven terraced agro-cultural heritage sites in China, but their perception of ecology and culture is not clear, and the polarization of different perception types remains prominent. There are two possible reasons for this. On the one hand, tourists' attitudes toward tourism determine, to a certain extent, their perception of a destination, and tourists who travel for leisure and entertainment attach great importance to the quality of the landscape and service experience at the destination. The beauty of scale and the lines presented by the physical landscapes of forests, villages, and terraces and the tangible tourism experiences such as accommodation and food are more likely to trigger tourists' intuitive feelings. The degree of landscape and service perception is thus extremely high. On the other hand, recreational purposes weaken the perception of the deeper non-material

ecological and cultural attributes of a destination and can even be neglected from the very beginning of tourism activities. At the same time, the variability of tourists' cognitive level and values may affect their perception of immaterial aspects of terraced agro-cultural heritage tourism sites.

The high frequency of the term "terraced fields" shows its importance, and the attention tourists pay to the agro-cultural heritage of the terraces when considering the seven subject sites. The significance of the scale and productive attributes of the terraces is reflected in terms such as spectacular, scale, and rice. There is also a side reaction that the development of terraced agro-cultural heritage tourist sites should maintain and increase the significance of the terraces as the main landscape form while ensuring that the aesthetic quality of the landscape and its ecological function are not reduced, and the agricultural attributes of the terraces are not lost. The degree to which tourists perceived the ecology and culture of the sites was extremely low, showing their lack of attention to these aspects of the subject sites; in response to this phenomenon, the management department could consider the proximity and concretization of the immaterial perception in the landscape up close, broadening how terraced farming culture is promoted in regional tourism, increasing education and practice programs related to ecological civilization, and enriching the ecological and cultural perception channels for tourists to these sites.

Tourists have a high preference for traditional terraced agricultural landscapes [46], which will contribute to the development of agricultural heritage landscape conservation policies [47]. Unlike other studies on tourists' perceived attitudes in terraced landscapes, which do not systematically classify perceived attitudes into different types, this paper presents a more comprehensive and systematic classification of tourists' attitudes in terraced agricultural cultural heritage, and on this basis, we conducted an in-depth analysis of the given types. The analysis results are beneficial for managers and policy makers of terraced agricultural cultural heritage to develop scenic strategies quickly and purposefully. At the same time, this idea and method can also provide reference suggestions for the analysis of tourists' attitudes and the development of optimization strategies for other thematic scenic spots.

This article considered important terraced agro-cultural heritage landscapes in China and collected rich research data from noted travel websites in China; however, the limitations of the study population remain, and differences in aesthetic preferences of different populations may have an impact on perception results. Additionally, there are objective differences in the number of people of different age groups using the Internet and writing online travelogues. The publishers of domestic online travel notes mostly fall within the youth and adult groups, and thus the results for the landscape perception, to a certain extent, lack universality and have objective cognitive limitations. The next step will be to analyze the differences in landscape perception of people of different age levels and education and to propose targeted landscape optimization strategies based on the results of this analysis.

## 6. Conclusions

Consideration of the internal factors affecting tourists' perceptions of terraced agro-cultural heritage sites is of great significance for improving their landscape quality, optimizing tourism services, and expanding regional ecological and cultural advantages. The results of this study yielded three main conclusions. First, tourists' perceptions of terraced agro-cultural heritage tourism sites are rich in content, and the summary of their attitudes reveals that they tend to focus on four objects of perception: landscape, ecology, culture, and service, therefore, the government and scenic area managers could focus on these four areas for policy and scenic area optimization strategy development. Second, there are significant differences in tourists' perception of the target sites, with the perceptions of landscape being the strongest (word frequency, 148,151, accounting for 47.11%) and ecology being the weakest (word frequency 16,793, accounting for 5.34%). Perception of service (word frequency 129,707, accounting for 41.25%) and culture (word frequency 49,818, accounting

for 6.30%) fell between. It is clear that tourists' perceptions of the site related to ecology and culture were extremely low. Thus, it is necessary to strengthen the protection of the physical landscape such as "terraces" and "architecture" and the improvement of tourism services, as well as to pay more attention to ecological and cultural aspects of terraced agricultural cultural heritage landscape. Third, tourists' perceptions of the attractions showed a highly positive emotional tendency and were dominated by highly positive and neutral emotions, accounting for 52.82% and 36.45%, respectively, while negative emotions accounted for only 10.73%, most of which (78.95%) were mild and only 2.16% were severely negative. The majority of tourists have a positive attitude toward the terraced agricultural cultural heritage landscape, which means that it has important tourism potential and can bring new development opportunities for rural tourism as a tourism resource.

**Author Contributions:** Conceptualization, X.Y. and Y.Z.; methodology, X.Y.; software, X.Y.; validation, J.Z., B.D. and C.S.; formal analysis, X.Y.; investigation, X.Y.; resources, X.Y. and Y.Z.; data curation, X.Y.; writing—original draft preparation, X.Y.; writing—review and editing, Y.Z.; visualization, J.Z.; supervision, C.S.; project administration, B.D.; funding acquisition, Y.Z. All authors have read and agreed to the published version of the manuscript.

**Funding:** This research was funded by the National Natural Science Foundation of China (No. 41771115).

**Data Availability Statement:** Data can be provided by the corresponding author upon request.

**Acknowledgments:** The authors would like to thank the National Natural Science Foundation of China (No. 41771115) for providing financial support for this research. I would also like to thank my research lab mates and dormitory roommates for their help in writing my thesis. We are grateful to the reviewers and editors for their suggestions on this paper, which were very constructive and helpful in improving the quality of the article.

**Conflicts of Interest:** The authors declare that they do not have any conflict of interest.

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
