# Peer review of "Tourists’ Perceived Attitudes toward the Famous Terraced Agricultural Cultural Heritage Landscape in China"

_agriculture, doi:10.3390/agriculture12091394_

Round 1

Reviewer 1 Report

Greetings,

The paper is well written, but some corrections need to be made. In the summary, it is necessary to write the results of the research. In the introduction, it is necessary to emphasize the objectives of the research and the contribution of the research. At the end of the introduction, it is necessary to briefly write what will be written about in the paper selections. After the introduction, a selection of the literature review should be written. When you write this selection, you will also strengthen the references of this paper because the number of references is small. References that will be in this selection should be younger than five years. In the Materials and Methods section, it is necessary to explain first how the research will be conducted, and only then do these sub-selections go. The results are well written, I have no complaints about them. As far as the selection of discussions is concerned, it is necessary to indicate the references in which similar research was done and to compare those results with the results obtained in this research. In the conclusion, it is necessary to state the limitations of the research and provide guidelines for future similar research.

All the best.

Author Response

Dear Reviewer:

    Thanks very much for your reviews’ comments concerning our manuscript entitled “Tourists' Perceived Attitudes Toward the Famous Terraced Agricultural Cultural Heritage Landscape in China” (ID: agriculture-1876714). The comments will be of great help in improving the quality of the paper. According to your advice, we have revised the manuscript extensively. Please see the attachment, and find my revisions in the re-submitted files. If there are any other modifications we could make, we would like very much to modify them and we really appreciate your help.

    Best wishes!

    Yours sincerely.

    Authors

Reviewer 2 Report

For me it is a pleasure to revise your interesting paper. The abstract is well written as the introduction. It presents the problem and the purpose of the study. Concerning the introduction, I suggest to present the structure of the paper in the last paragraph of it. 

In relation to section 2 – Materials and methods – I suggest to present in the subsection 2.2. the period to which it refers “the travel diaries posted by tourists on travel websites”.

In relation to Results and Discussion of them, I also think that these sections are well written. For me the weak section is the Conclusion one. Please relate your conclusion with literature review.

Also present here the political or practical implications of the study (for me this is very important). Last but not the least add the limitations of the study and paths for future research.

Good work on the next steps!!!

Author Response

(The authors gave the same response as above.)

Reviewer 3 Report

The topic and the methods used are getting more and more common in articles prepared by Chinese scientists. That is why I belive that the article must be further imporved, since the results should lead to specific conclusions and recommendations for policy-makers, tourist businessmen etc.

In my opinion, the abstract must include the research objectives as well. All the chapters must be developed to provide more details about the background, the literature, the results and conclusions. More literature must be also referred in the article to make it more justified.

Author Response

(The authors gave the same response as above.)

Reviewer 4 Report

Dear authors,
Thank you for sharing your research findings with us. I find your paper "Tourists' Perceived Attitudes Toward the Famous Terraced Agricultural Cultural Heritage Landscape in China" interesting and well written. The language of the text is pleasant and easy to understand. Both the methodological part and the results are solid. I would have few suggestions for the authors to:
- emphasise in the abstract the scientific contribution of the research
- add a brief literature review to support the scientific gap
- If possible, add information about the time frame of the data collected on the websites (e.g., the last 5 years including 2022, or 3 years to 2022...)
- In the concussion / discussion part, the managerial and theoretical implications as well as the identified limitations of the research.
Good luck.

Author Response

(The authors gave the same response as above.)

Reviewer 5 Report

Thank you for a chance to read this interesting article. There are few things that I would like to point to: 

Introduction part needs to strengthen by adding current and relevant information, since the latest relevant reviews are not incorporated. The aim of the research needs to be more clear presented and the reasons why this kind of research is necessary. Discussion: The authors summarized the results obtained from the preceding analysis. However, I would like to ask whether the authors have found anything different from the previous research. It would be more interesting if the authors would indicate some differences and state the possible reasons for such differences. The inclusion of such differences would make this research much more interesting and meaningful. - Section conclusions does not bring anything new, authors just repeated presented results. Please, rewrite this section by adding theoretical and practical implications, limitations of research and recommendations for future research. 

Author Response

(The authors gave the same response as above.)

Round 2

Reviewer 1 Report

Greetings,

The authors corrected the paper in accordance with the reviews. The paper should now be accepted.

All the best.

Reviewer 5 Report

Authors paid good attention to the reviever's comments and I believe that the revised version is much improved. I suggest publishing this paper.